# Current State and Innovations in Newborn Screening: Continuing to Do Good and Avoid Harm

**DOI:** 10.3390/ijns9010015

**Published:** 2023-03-17

**Authors:** Giancarlo la Marca, Rachel. S. Carling, Stuart. J. Moat, Raquel Yahyaoui, Enzo Ranieri, James. R. Bonham, Peter. C. J. I. Schielen

**Affiliations:** 1Newborn Screening, Clinical Chemistry and Pharmacology Lab, IRCCS Meyer Children’s University Hospital, 50139 Florence, Italy; 2Department of Experimental and Clinical Biomedical Sciences, University of Florence, 50139 Florence, Italy; 3Biochemical Sciences, Viapath, Guys & St Thomas’ NHSFT, London SE1 7EH, UK; 4GKT School of Medical Education, Kings College London, London SE1 1UL, UK; 5Department of Medical Biochemistry, Immunology & Toxicology, University Hospital Wales, Cardiff CF14 4XW, UK; 6School of Medicine, Cardiff University, University Hospital Wales, Cardiff CF14 4XW, UK; 7Laboratory of Metabolic Disorders and Newborn Screening Center of Eastern Andalusia, Málaga Regional University Hospital, Institute of Biomedical Research in Malaga (IBIMA-Plataforma BIONAND), Avenida Arroyo de los Angeles s/n, 29011 Malaga, Spain; 8Biochemical Genetics, Genetics and Molecular Pathology, SA Pathology, Women’s & Children’s Hospital, Adelaide 5043, Australia; 9Sheffield Children’s NHS Foundation Trust, Western Bank, Sheffield S10 2TH, UK; 10International Society for Neonatal Screening, Reigerskamp 273, 3607 HP Stichtse Vecht, The Netherlands

**Keywords:** neonatal screening, next-generation sequencing, biochemical analysis, SCID screening, cystic fibrosis screening, proteomics, metabolomics, quality control, 2nd-tier testing, NGS in NBS

## Abstract

In 1963, Robert Guthrie’s pioneering work developing a bacterial inhibition assay to measure phenylalanine in dried blood spots, provided the means for whole-population screening to detect phenylketonuria in the USA. In the following decades, NBS became firmly established as a part of public health in developed countries. Technological advances allowed for the addition of new disorders into routine programmes and thereby resulted in a paradigm shift. Today, technological advances in immunological methods, tandem mass spectrometry, PCR techniques, DNA sequencing for mutational variant analysis, ultra-high performance liquid chromatography (UPLC), iso-electric focusing, and digital microfluidics are employed in the NBS laboratory to detect more than 60 disorders. In this review, we will provide the current state of methodological advances that have been introduced into NBS. Particularly, ‘second-tier’ methods have significantly improved both the specificity and sensitivity of testing. We will also present how proteomic and metabolomic techniques can potentially improve screening strategies to reduce the number of false-positive results and improve the prediction of pathogenicity. Additionally, we discuss the application of complex, multiparameter statistical procedures that use large datasets and statistical algorithms to improve the predictive outcomes of tests. Future developments, utilizing genomic techniques, are also likely to play an increasingly important role, possibly combined with artificial intelligence (AI)-driven software. We will consider the balance required to harness the potential of these new advances whilst maintaining the benefits and reducing the risks for harm associated with all screening.

## 1. History and Current State of NBS

An overview of newborn screening (NBS) has been presented in several reports, e.g., on Europe [1] South-East Asia and the Pacific [2], the USA [3], and Latin America [4]. The most recent global overview was published in 2015 [5].

Many of these report the similarities and differences between screening programmes, e.g., the average number of samples per laboratory, the time between birth and sampling or sampling and analysis, the percentage of babies screened compared to the birth rate, and whether parental consent for participation or sample storage is sought. Despite these important ethical and practical differences, attention is often focused on the number of disorders included in any program as a hallmark of improvement and quality in each country or region. In Europe [1], of 51 countries, 47 screen for PKU, 46 for congenital hypothyroidism (CHT), 25 for cystic fibrosis (CF), 24 for congenital adrenal hyperplasia (CAH), and 7 for severe combined immunodeficiency (SCID). While screening for more conditions does not always result in a more effective program, the perceptions of public opinion, family advocacy groups, and, sometimes, political attention may focus largely on this aspect.

Beginning in 1963, analytical methods were developed and employed to screen for PKU, CHT, and CAH. To quantify phenylalanine in dried blood spots (DBS), Guthrie developed a low-cost test with high throughput and a rapid turnaround that would enable newborn screening. He used a bacterial growth inhibition (β-2-Thienylalanine) assay [6] that was adopted globally, but perhaps his most significant contribution was to popularize the use of a filter paper card to collect a small drop of blood from a baby’s heel, allowing for easy transport to a testing laboratory—this became known as the dried blood spot (DBS) or ‘Guthrie’ card. This DBS sample is used to this day for analyses in NBS laboratories around the world. During the 1980s and 1990s, microbiological inhibition and, sometimes, thin-layer chromatographic assays were gradually replaced by colorimetric and fluorimetric techniques providing greater analytical sensitivity. In the 1970s and 1980s, CHT was added to neonatal screening by many screening panels [7]. Measurements of thyroxine (T4) along with beta-microglobulin were used to screen for CHT. This was soon replaced by the measurement of thyroid stimulating hormone (TSH) using a sensitive (radio-) immunometric assay. In some programmes, the measurement of T4 was also used with TSH to identify a larger proportion of hypothyroidism (hypopituitary). Another endocrine disorder, congenital adrenal hyperplasia (CAH), was subsequently added in many countries using the measurement of 17-hydroxyprogesterone (17-OHP). The initial assays for 17OHP were performed using a (radio-) immunometric method and were followed by more sensitive, time-resolved immunoassays [8]. With the introduction of CHT and CAH screening, a practice was introduced to sample a neonate a second time, e.g., a week after the first blood sample was taken, to clarify the significance of an initial out-of-range result.

Shortly after the inclusion of these disorders, newborn screening for cystic fibrosis (CF) by measuring immunoreactive trypsinogen (IRT) through immunoassay was introduced in some countries. The development of the IRT assay was credited to the work of Janet Crossley [9] who showed the correlation between elevated IRT in babies with CF. This was followed by the development of several two-tier screening strategies using a combination of IRT with the measurement of pancreatitis-associated protein (PAP) or limited DNA analysis, as a second-tier test performed in a subset of those samples with an elevated concentration of IRT—usually the top 1%. These ‘second-tier’ approaches were used in combination to enhance both sensitivity and specificity [10,11,12,13].

In the late 1990s and early 2000s, multiplex metabolite techniques were pioneered by Chace and Millington [14], to analyse over 60 small molecules in a single analytical run using tandem mass spectrometry (MS/MS). This changed the accepted paradigm of NBS from ‘one spot—one test—one disorder’ to ‘one spot—one test—many metabolites for many disorders’, resulted in a quantum change in how screening was undertaken to identify inherited errors of metabolism (IEM) [15], and led to the development of screening panels ranging from 10–25 disorders in some countries.

By 2010, all these developments led to the first attempt to make a full overview of analytes used globally to screen for disorders in NBS. By combining all available data in both normal populations and confirmed positive cases from many laboratories around the world, data from 11,000 NBS disorders became available [16,17]. No individual regional or national program could achieve this in isolation, thus underlining the immense importance of international collaboration in the study of rare disorders and the field of NBS. This is an outstanding achievement for the team that undertook the collation and statistical interpretation of global data to develop clinical interpretation algorithms.

Furthermore, in 2010, the quantification of T-cell receptor excision circles (TRECs) to screen for SCID introduced the use of front-line DNA-based technology as a primary NBS test by quantifying TRECs using a PCR-based technology [18]. Most recently, screening for SMA has been introduced into screening programmes in some regions and countries [19,20] as the first disease that does not have a biochemical (or TREC-number-based) footprint in blood. Instead, screening is undertaken by determining mutations of the SMN1 gene, specifically exome 7 deletion, in the baby’s genome, thus making it the first example of a first-tier genetic screening test. With these advances in DNA-analysis-based techniques, which today are often next-generation sequencing (NGS)-based, it is clear that 2023 is set to see the development of population-based screening using large panels of mutations or whole exome or genome sequencing (WES or WGS). There are several research-based programs underway to determine whether NGS can be used in an NBS setting. This includes Genomics England, which will be applying NGS to more than 100,000 babies to identify a pre-selected group of treatable disorders. Other programs in Europe, the USA, China, and Australia are also proceeding to use NGS in WES or WGS in NBS.

The last sixty years, from PKU to WGS, have provided both opportunities and challenges. It is firmly recognised that ‘All screening programmes do harm, some do good as well, and, of these, some do more good than harm at reasonable cost’ [21]. This important challenge remains a driver to continually improve both the specificity and sensitivity of testing, seeking to ensure that families are not harmed by an unexpected false-positive result or left with significant anxiety due to a result of uncertain significance while, at the same time, ensuring that true-positive cases that would benefit from early and effective treatment do not escape detection. Below, we will address the innovations and quality assurance measures accompanying those innovations to improve these safeguards and prepare NBS for the future.

## 2. Good (Laboratory) Practice and Reference Materials

Effective quality assurance includes pre-analytical, analytical, and post-analytical aspects—all influencing the quality of the result that is reported for clinical action. Internal quality control (IQC) has always been central in the NBS laboratory, particularly as regional screening laboratories in the same country adopt nationwide cut-off values (COVs). IQC also needs to evolve in parallel with the introduction of new techniques. To this end, an overview of recent developments and insights into the IQC of NBS that are necessary to maintain the high standards of the past decades is provided below.

### 2.1. Volumetric Standardization

The effects of blood volume applied to the collection device, haematocrit (Hct), and spot size all impact the quantitative result obtained [22,23]. Furthermore, when NBS results are quantified by stable isotope internal calibration (SIIC) alone, which is routine practice for many screening programmes, the volume of blood assumed to be present in the sub-punch directly impacts the quantified result. The volume was previously reported by the CDC to be 3.1 µL [24]. Recent work, based upon a gravimetric approach [25] and a direct comparison with a volumetric collection device, has, however, contraindicated this [26]. Comparisons of experimental data from various studies are further complicated by the different types and grades of filter paper used in the collection devices. Volumetric standardization will remain a key issue in the routine screening and evaluation of new methodologies. 

### 2.2. Calibration

An issue common to all screening methods, but especially MS/MS, is calibration. In MS/MS, some laboratories quantify results using stable isotope dilution and a DBS calibration curve, while others use SIIC alone. In principle, the former approach benefits from a matrix-matched calibration curve and the assurance of linearity across the defined range of concentrations. However, the practicalities of accurately preparing multiple DBS calibrators in sufficient quantity pose significant challenges, and this is before the traceability and standardization of the calibrators are considered. The alternative, more practical approach, is to rely on SIIC; however, the efficacy of this is limited by the purity of the stable isotope internal standards (IS), the accuracy with which they are prepared, and the limitations of what is, effectively, a single-point calibration [27]. With the introduction of proteomics and metabolomics profiling in screening and the rapid increase in the number of new biomarkers, the appropriate calibration of individual markers may prove to be challenging.

### 2.3. Clinical Utility and the Application of COVs

Although accuracy is an important factor, the clinical utility of a screening test is determined by the evidence base associated with the COVs. In countries that must adhere to nationally agreed-upon protocols based on single analyte COVs, inter-laboratory harmonization is critically important, as is the standardization of the assay from which the COVs were originally determined. With numerous laboratories relying on in-house methodologies, harmonization and standardization are challenging. Currently, there are no commercially available, DBS-based external calibrators. Recent studies have demonstrated that inter-laboratory variation can be significantly reduced using a common stable isotope [28] and by retrospectively harmonizing screening results to a common material [29].

### 2.4. Production of Certified Reference Materials

A major limitation of the utility of DBS specimens for biomarker analysis is the lack of commercially available, matrix-matched certified reference materials (CRMs) for the various analytes in DBS specimens with which to standardise laboratory tests [27,28]. As a result, DBS calibrators tend to be produced in-house by collecting blood from a healthy donor or using residual pooled patient samples and adding an aqueous enrichment prior to application onto the filter paper. The exact preparation of the DBS calibrator varies between laboratories, e.g., the volume of blood added to the filter paper, the type of filter paper, the varying Hct of the specimen, or the use of lysed blood specimens. Each factor can affect the measured concentration. Importantly, the analytical method used to assign the DBS calibrator values can also influence the accuracy of the analytical result. The consistent and accurate preparation of DBS calibration curves can also pose a challenge to laboratories by impacting imprecision.

### 2.5. Precision of DBS Assays

The precision of the DBS assay should be assessed using appropriate DBS IQC specimens prepared at a minimum of three different concentrations (low, intermediate, and high), IQC materials provided by reagent manufacturers alone should not be relied upon in this context, and ‘third-party’ IQC materials should be obtained or produced. COVs should be considered when determining the optimum IQC concentrations. It is good laboratory practice to prepare IQC samples and calibrators using whole blood from a healthy individual in order to match the matrix of a newborn blood spot specimen. The specimens should be at an appropriate haematocrit (Hct) level that is within the range of the patient group. Furthermore, it is important that the endogenous concentration of the analyte being tested is at a known level and that there are no iatrogenic factors, such as drugs or parental nutrition (intralipids) that may result in interferences to the analytical tests.

The intra- and inter-assay precision, from the replicate analyses, should ideally be set at <15% for dried bloodspots [30]. However, achieving the appropriate and reproducible extraction recovery of a screening analyte from a DBS can be challenging and will vary from one screening analyte to another. Appropriate validation experiments are required to optimise analyte recovery from DBS specimens. The recovery of the analyte need not be 100% but should be precise, especially where the absolute values of analytes are used in algorithms and where the results in affected infants may be close to the screening COVs. Clear QC rules should be in place to determine batch acceptance, identify out-of-range performance, and reveal trends in results over time.

### 2.6. External Quality Assessment (EQA) Schemes for Comparisons of Performance between Laboratories

It is recommended that all laboratories offering DBS biomarker analysis should participate in EQA schemes or sample exchange(s) when an EQA scheme is not available, as it is an essential tool for ensuring that performance is comparable to that of other laboratories. However, laboratories must also be aware of the limitations of these schemes, e.g., the limited number of participants, the method groups, the results reported relative to the all laboratory trimmed mean, the spiked value, the use of multiple analytes, the inclusion of multiple methods but only a single assessment, and also the limited number of sample distributions.

### 2.7. Monitoring of Population Data

Due to the limitations of sourcing appropriate IQC material, laboratories often elect to monitor population data using statistical procedures. Such population statistical monitoring provides the advantages of improved confidence in the precision and accuracy of the analytical method over longer time periods. Population parameters, such as the median, multiples of the median (MoM), and percentiles, will provide a measure of statistical significance that enables a direct comparison with other laboratories in the screening programme and the improved detection of any outliers or trends. The disadvantage of this approach is that it can only be applied retrospectively. A more sophisticated solution would be to adopt patient-based, real-time quality control (PBRTQC). PBRTQC is the statistical analysis of results in real-time from a defined patient population to identify bias, shifts, and drifts in analytical performance [31,32,33,34]. There are very few, if any, other biochemical tests that are performed on such a defined and consistent population; hence, NBS lends itself perfectly to PBRTQC. Results generated from a rolling screened population mean should not vary unless there is an analytical error or a change in the screened population, with the latter being a very rare occurrence. By analyzing moving averages, PBRTQC can detect very small changes in the bias of a given assay in real-time. Other benefits include the ability to assess the pre-analytical phase as well as the analytical; reduced reliance on commercial IQC materials, which are often expensive and of limited utility; the negating of any commutability issues; reduced reliance on in-house IQC; and improved quality of NBS results.

The expected future advances in the development of new techniques and methods will impart greater demands on the need for rigorous IQC, especially NGS analysis and possibly proteomic and metabolomic methods. It is evident that many of the IQC mechanisms and materials will need to be developed in parallel with the introduction of these new techniques. The CDC has now added proficiency testing schemes for TREC and SMN1 to their services (https://www.cdc.gov/labstandards/pdf/nsqap/2019_Q4_TRECPT_Report-508.pdf—accessed on 12 March 2023). 

## 3. A Putative Role for Second-Tier Screening

NBS typically applies inexpensive tests performed on an entire newborn population. The last two decades have seen a rapid expansion in the number of disorders in screening panels and the introduction of screening tests that do not have the desired specificity to keep the number of screens at increased risk (screen-positive results) adequately low. This has resulted in the increased adoption of laboratories performing higher-specificity, second-tier testing on the initial specimen that produced a presumptive positive. These second-tier tests, (often analytically complicated, more expensive, and with lower throughput) have been used to significantly reduce the number of false positive results generated by the first-tier primary test, thus reducing the need for clinical referral.

There are several reasons why first-tier tests are not sufficiently specific. The specificity of immunoassays can be compromised by the cross-reactivity of antibodies with other related, non-specific targets (17-alpha-hydroxy-progesterone) [35,36]. In mass spectrometry, false-positive test results are due to the presence of isobaric interferences—i.e., compounds with the same mass-to-charge ratio (*m*/*z*) but that are structurally different—with examples being leucine/isoleucine vs. hydroxyproline and isovalerycarnitine vs. pivaloylcarnitine vs. 2-methylbutyrylcarnitine [37,38]. Specificity also depends on using derivatized vs. non-derivatised methodology in MS/MS. Derivatization by butylation is still used by laboratories because the formation of the butyl-esters “quenches” the negative charge on the acid moiety of amino acids, and acylcarnitines result in an increase in both ionization efficiency and sensitivity. Technological improvements in MS/MS instrumentation have resulted in increases in sensitivity, thus allowing for the detection of amino acids and acylcarnitines as their native, underivatized, free acids. This has reduced the sample preparation time and circumvented the use of corrosive butanol-HCl. The shortcomings of this non-derivatized method are the inability to differentiate between isobaric acylcarnitines (mainly C3DC/C4OH and C4DC/C5OH) and a reduction in sensitivity for measuring low-mass amino acids (e.g., glycine and alanine). For these reasons, some laboratories choose to continue to use the derivatization technique as a second-tier approach. 

Second-tier testing—for example, employing a chromatographic separation—can improve the specificity and the diagnostic performances of the screening test [39].

An example in MS/MS-based NBS for selected inborn errors of metabolism (IEM) is screening for methylmalonic aciduria and propionic aciduria, for which propionyl-carnitine (C3), along with low levels of methionine, is used as the primary marker. The introduction of a second-tier test to detect the level of methylmalonic acid (MMA) from the initial bloodspot has vastly improved the screening test for propionic aciduria and methylmalonic aciduria, resulting in the reduction of the unnecessary recalls associated with this biomarker [12,13].

Many countries have already adopted the use of genetic, second-tier DNA tests as a part of their NBS programmes [40,41], and for some conditions, this has shown a great benefit by excluding healthy carriers or pseudo-deficient individuals.

Despite these benefits, there are still unresolved issues involved in the use of second-tier, DNA-based testing. Greater turnaround times associated with DNA-based testing may result in delays in the recall of an infant suspected of having an acute metabolic decompensation disease. Furthermore, there remain vigorous debates about the efficacy of the use of genetic variants identified during the course of newborn screening towards clinical or solely biochemical significance. The occurrence of variants of uncertain significance (VUS) can limit the clinical utility of genetic findings in a screening context. (see also Section 6 ‘Genetic Methods in NBS’).

Recently, an up-to-date review was published on the combined application of high-pressure liquid chromatography (LC) and MS/MS, both as a first-tier or second-tier technique, for more than 20 disorders (particularly lysosomal storage disorders) [42]. This combination can offer significant benefits in some situations: (1) It cleans up the samples that are injected in the MS/MS. (2) Using LC, the analytes are concentrated in a smaller volume and, therefore, a higher concentration. (3) Isobaric interfering substances can be separated from the analytes of interest prior to analysis in the MS/MS. (4) Finally, for the MS/MS measurement of the products of enzymes, the LC-mediated separation of the substrate and product prevents interference by enzymatic activity within the produced product, with the substrate being degraded by the MS/MS measurement itself. With the continued development of analytical platforms, the known disadvantages of LC-MS/MS (e.g., the cost of equipment, time per analysis, and maintenance, to name a few) are presenting less of a barrier.

## 4. Proteomics/Metabolomics

Metabolomic and proteomic technologies have emerged as complementary strategies to genomics in an attempt to understand human diseases. These approaches have developed rapidly in the past years and offer the possibility of discovering potential diagnostic, therapeutic, prognostic, and patient stratification biomarkers.

Metabolomics involves the identification and quantification of metabolites present in a biological system. For metabolomic analysis, sample preparation is a critical process and will depend upon the type of metabolites to be studied and the platform to be used. Mass spectrometry-based metabolomics is the most extended approach, where MS is generally combined with gas chromatography or LC online. Metabolomics for measuring metabolites of <2000 amu, using UPLC time-of-flight (TOF) to provide an untargeted approach, has made significant advances in identifying new and novel markers—referred to as features—for the potential to screen for an increasing group of disorders. An example of this is the use of untargeted lipidomics to identify novel lipids for the screening of peroxisomal disorders (PD). The current screening for a specific PD, X-ALD, utilizes the measurement of lyso-C26:0-phosphatidylcholine from dried bloodspots. This single marker, whilst having good sensitivity, lacks specificity. The use of an untargeted UPLC-QTOF analysis generates a large number of features (thousands), each with assigned mass-to-charge ratios (*m*/*z*)—in both positive and negative ion modes—and retention times. Identifying those features that are informative requires the use of high-end machine learning statistical methods followed by the analysis of predictive testing. This approach has identified a number of additional lipid markers with the potential for screening for more PD [43].

Metabolomics has also been employed in DBS to identify potential early biomarkers of biliary atresia [44] and autism [45]; predict, at birth, the clinical phenotype of some diseases, such as cystic fibrosis [46,47] and very long-chain acyl-CoA dehydrogenase deficiency [48]; or retrospectively assess the early-life exposome [49]. It is noteworthy that the metabolomic status of neonates depends upon the metabolomic profile of their mothers at the end of pregnancy [50] and may be influenced by nutrition [51], gestational age, and age at the time of specimen collection [52].

Proteomics involves the large-scale study of the entire protein complement of a cell, tissue, or organism under a specific set of defined conditions, and includes both their structure and physiological functions. Many techniques are used to identify proteins as reliable biomarkers, such as tissue arrays or mass spectrometry. All of the approaches include three essential steps: extraction and separation, identification, and verification of proteins or peptides. A two-step approach of MS/MS is first used to acquire the data necessary to identify individual proteins or peptides, followed by bioinformatics to analyse and assemble the MS/MS data. Advances in technology have enabled thousands of proteins to be resolved, and as in metabolomics, proteomics is critically dependent on bioinformatics to process the raw mass spectral data into protein data and map them onto the cellular structure and functions.

In the last decade, the applicability of these approaches in NBS has been evaluated. Some proofs of concept for newborns have been reported, which are usually performed with DBS specimens but also with dried urine spot specimens [53,54].

To date, most proteomic-based applications in DBS have been used to quantify target proteins to screen for five primary immunodeficiency syndromes (X-linked agammaglobulinemia, Wiskott–Aldrich syndrome, X-linked chronic granulomatous disease, DOCK8 deficiency, and ADA deficiency), using signature peptide biomarkers [55,56,57]. A method, based on an immunoassay platform targeting 22 proteins, may also detect 22 innate immunodeficiencies affecting the complement system and respiratory burst function in phagocytosis in DBS [58]. Recently, MALDI-TOF has been used to screen for haemoglobinopathies from dried blood spots [59].

Despite these advances, it remains to be seen whether metabolomics and proteomics techniques will provide a useful, practical way for the NBS laboratory to detect new conditions, reduce the number of false positives, and/or discriminate between likely pathogenic phenotypes or those newborns that may benefit from close monitoring.

## 5. Artificial Intelligence and Machine Learning in NBS

Another post-analytical approach to improving the quality of NBS results would be to adopt a more sophisticated approach to data review and interpretation. Many laboratories still rely on individual screening algorithms for each disorder, which are typically based on the concentration of a single analyte or the ratio of the primary analyte to a non-condition specific analyte, e.g., octanoyl:decanoyl carnitine for medium-chain acyl-CoA dehydrogenase deficiency screening. There has been a paradigm shift, due to the amount of data that can be generated from a single dried bloodspot, where no longer using a discrete test but profiling provides the optimal identification of disease. One such interpretative system is that of the Collaborative Laboratory Integrated Reports (CLIR), which makes use of multivariate analysis to generate a discriminant score reflecting the likelihood of a diagnosis based on condition-specific disease ranges for all informative analytes [17,60]. The customisation of these tools is provided for individual laboratories to correct for differences in analyte panels and sample preparation. Access to the algorithm is free of charge and available on demand, enabling these interpretative tools—which have been shown to reduce false-positive rates and improve positive predictive values—in real time.

With databases on rare diseases, genetic databases, and screening data (including matching demographic data for stratifying results, e.g., date of sampling, birth weight, method of analysis, etc.) becoming more interconnected, and algorithms becoming more powerful, we expect these automated algorithms to become increasingly reliable in their predictions and find greater application in routine NBS in the future.

One very interesting development is the Enhancing Data-driven Disease Detection in Newborns (ED3N) pilot program. The ED3N program attempts to assess the functionality of a CDC data platform in the Division of Laboratory Sciences Newborn Screening and Molecular Biology Branch. The project aims to help improve disease detection in newborns through DBS screening employing both biochemical and molecular analytical approaches. It is anticipated that the use of both biochemical results to provide phenotypic information and genomic data could greatly improve and extend the disease prediction tests that are currently available (https://www.cdc.gov/nceh/dls/ed3n.html—accessed on 12 March 2023).

## 6. Genetic Methods in NBS

The potential for the use of rapid and less-expensive genetic analysis to change screening was first recognized in 2008 when Andermann et al. revisited the Wilson and Jungner in the genomic age [61]. Interestingly, this was before the introduction of massively parallel sequencing, which revolutionized genomic analytical methods. The technique of massively parallel sequencing [62] was applied to a small sample of DNA isolates from amniocenteses, identifying trisomy 21, 18, and 13 among samples of unaffected fetuses, suggesting an application in prenatal screening. Since 2008, the technique itself has continued to be developed and is in its fifth innovative generation—now generally referred to as next-generation sequencing (NGS); it has found applications in many fields of medicine, including NBS. Other DNA-based analytical methods—from micro-arrays to line blots, MLPA, and various PCR techniques—have also continued to develop, making them more accessible and offering the potential for applications in screening.

DNA analysis can increasingly play a role as a first-line or second-tier analytical technique in a screening context. As a first-line test, multiple approaches are possible, including targeted screening; focusing on a gene or a panel of genes to identify mutations in those genes with defined pathogenicity; or as a non-targeted approach, where, principally, the whole exome or whole genome of the neonate is analysed [63].

Until 2010, DNA analysis fulfilled a modest, second-tier role—mostly limited to CF screening with the analysis of the CFTR gene variants—initially using reverse hybridization on commercial line blot assays [11]. That changed with the introduction of SCID screening in NBS in the United States, where a quantitative PCR method was used to quantify TRECS in DBS. This laboratory-developed test became a commercial product that is now currently offered by several companies. The TREC assay became the first DNA-based molecular test to be applied as part of newborn screening at the population level [64]. The recent introduction of SMA screening, relying upon the detection of deletions in the SMN1 gene, is the first example of a whole population test that targets specific mutations causing the disorder that is to be identified. Thus, for the first time, a monogenetic disorder with no biochemical footprint in the blood can be detected in NBS, paving the way for many other conditions. Both SCID and SMA screening are developing dynamically, with SMA screening introduced in 2021 and 2022 as part of national screening programs in certain areas, such as Norway, Germany, and New South Wales (Australia), and regional pilots underway in many other countries; SCID screening was recently introduced in the Czech Republic. As part of SCID screening, the possibilities of screening for X-linked agammaglobulinemia, accompanied by low or absent numbers of Kappa-deleting recombination excision circles (KREC) in DBS, are explored. These developments are stimulated by the commercial availability of multiplexed laboratory tests for TREC, KREC, and SMN1.

Taking this concept one step further, with the current potential of multiplexing molecular tests, would enable the design of panels targeted towards a selection of IEM or a collection of related genes. An example is the use of qPCR to identify both SMA and SCID from a single assay. In a diagnostic setting, these panels have already been used [65,66]; however, they have not yet in a screening setting.

Perhaps the greatest potential is the ability to analyse the entire exome or even genome (WES or WGS), but this comes with risks, such as identifying carriers, late-onset disease presentation, and pseudodeficiencies. This concept of the use of WES and WGS was first published in 2013 [67]. Two large retrospective and prospective studies found that WES had an overall lower sensitivity and specificity when compared to routine NBS [68,69]. Their data clearly identified major challenges in this approach that need to be overcome before the implementation of NGS as a first-tier approach can be considered. Recently, two large, exploratory studies investigating the feasibility of the use of NGS in NBS were presented. One is Screen4Care, a European collaborative innovative medicines initiative, with a predicted 18,000 participants and relying on the use of custom-designed panels (https://screen4care.eu—accessed on 12 March 2023). The other is the Newborn Genomes Programme in the United Kingdom (https://www.genomicsengland.co.uk/initiatives/newborns—accessed on 12 March 2023), with a predicted 100,000 participants using WGS as a first-line test. The study design of this program has carefully agreed to reporting arrangements aimed at targeting serious treatable disorders that present during early childhood.

## 7. Safeguarding NBS in the Realm of Unbridled Technical Possibilities

Since its introduction in the 1960s, newborn screening has been adopted in most developed countries and it is estimated that, worldwide, more than 40 million babies are now being offered some form of newborn blood spot screening each year. While this includes the recent spread of limited screening for conditions, such as congenital hypothyroidism and sickle cell disease, in low- and middle-income settings, it still suggests that only around 30% of children born are offered any form of screening for easily treatable disorders. There is considerable interest in more widely extending the benefits of appropriate newborn screening if it can be conducted in a way that is ethical, safe, and cost-effective in healthcare settings where resources and access to treatment may be limited.

In most developed countries, newborn screening enjoys high uptake levels when offered, typically exceeding 99%—contrasting, sometimes sharply, with the uptake of vaccines offered during childhood.

Therefore, while new developments in screening technology, such as primary genomic testing, offer a great deal of potential, it is crucial that they do not undermine public confidence and the efforts of the past sixty years to develop acceptable and effective screening programmes. There is, for instance, some evidence that, when using approaches where the significance of the findings for the individual child may be unclear, public confidence regarding effective treatments offering benefits to the affected child is undermined, and this should be reflected in the design [70] and application of any new approaches to newborn screening.

While newborn screening has been a significant advance in public health, some areas need particular attention. One area is case definitions. NBS appears to be increasingly detecting patients with greater phenotypic heterogeneity. This makes the clarity of case definition based on agreed-upon confirmatory testing even more important. The development of agreed-upon case definitions and standardized treatment protocols would do much to enable and inform long-term outcome studies in patients identified by NBS, and these are often lacking despite programmes having been in place for many years. As technology offers new opportunities to extend the range of conditions identified, it is important that the care given to the selection of serious conditions leading to effective treatment is not relaxed and that clear case definitions with arrangements to assess the long-term outcomes and progress of screen-identified cases are in place before screening begins.

Finally, it is recognized that screening is not just simply a test but that the effective delivery of NBS rests upon the careful design and monitoring of the screening pathway from the delivery of pre-screening information to parents to entry into appropriate clinical care for those identified to have a disease [71].

These aspects, agreed-upon case definitions, long-term outcome studies supported by interoperable disease registries, and a clear understanding of the best way to conduct NBS and monitor the pathways are areas in which international collaboration is vital [72]. The conditions screened for are often rare, affecting much fewer than 1:2000 individuals, and even large countries may not have meaningful data, given the phenotypic heterogeneity of the patients identified.

In Europe, while professional Societies (such as the International Society for Neonatal Screening (ISNS) and the Society for the Study of Inborn Errors of metabolism (SSIEM)), and organisations for patient groups (such as EURORDIS), as examples, can support and stimulate this information sharing, collaboration at the governmental level is also needed. In recent years, stakeholder groups, such as Screen4Rare (www.screen4rare.org—accessed on 12 March 2023), have sought to develop a discussion and provide information to enable effective health policy development and inform political representatives, while recognizing and respecting national autonomy. 

As new developments in screening extend and challenge current thinking in this important area, there is a strong case for the cohesion offered by an expert advisory group—recognized by the European Commission—which can provide informed and unbiased advice and support health policymakers at a national level seeking to develop NBS as part of their national rare disease strategy.

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
