# Peer review of "Current State and Innovations in Newborn Screening: Continuing to Do Good and Avoid Harm"

_2409-515X, 2023, doi:10.3390/ijns9010015_

Round 1
Reviewer 1 Report
The manuscript entitled “Current state and innovations in newborn screening: continuing to do good and avoid harm” presents a high-level overview of the current state of newborn screening around the world. The authors highlight the ever-increasing list of disorders and biomarkers that are added to newborn screening panels, and different methodology approaches to improving the benefits of newborn screening while maintaining the ethical and cost-effective approaches that have made newborn screening such a successful and trusted public health program. Overall, it is a well-written manuscript, but I do have some minor suggestions below:
Line 82: Abbreviation CH should be CHT for consistency.
Line 122: Missing period at end of paragraph, WES acronym not yet defined.
Line 203: Making a statement that the bias from enrichment concentration values should be < 15% is not appropriate for DBS extracts. Many analytes are not going to have accuracy of > 85% for a DBS extract when using an SIIC approach rather than calibrators because the recovery from DBS plays such a large role. Succinylacetone, SUAC, for example is well-known to have recovery of less than 85% from dried blood spots. However, as long as the precision is within 15%, the low recovery and low accuracy is not necessarily problematic as the cut-offs can be set accordingly.
Line 252: The term “so-called” in the phrase “so-called second-tier tests” seems to give a skeptical tone that is not appropriate in this case. Suggested rephrasing: “This has resulted in the increased adoption of labs to perform higher-specificity second-tier testing on specimens that are presumptive positive following the first-tier screen.”
Line 252: Here, I believe it is important to mention that these second-tier tests are also often much lower throughput than first-tier assays. Someone less familiar with newborn screening may see higher specificity and lower false positives, and wonder, “why not just run that assay on all specimens in the first place?”
Line 257-261: This could be split in to 2 sentences, one discussing isobaric interferences, and the second mentioning the shift of many labs from derivatized methodology to non-derivatized approaches. This would allow you to possibly discuss that the non-derivatized approaches are attractive to labs, but no longer separate the isobaric pairs like C3DC/C4OH, and some labs actually use the derivatized approach as a second-tier method for these analytes.
Line 309: ‘Currently’ should be ‘Current’
Author Response
We would like to thank this reviewer for her/his valued work and compliments. Below, we have added a point-by-point rebuttal to her/his comments.
On behalf of the team of authors,
Sincerely yours,
Peter Schielen
Abbreviation changed to CHT (line 82)
Period added (line 122)
Abbreviation WES and WGS introduced (line 123)
We agree with this reviewer's observation and have revised the statement (line 203-206).
We have rephrased this sentence incorporating this reviewers suggestion, leaving out the term ‘so-called’ (line 254-256)
We added the aspect of higher through put times as a point in line 257-258.
We have adopted this reviewers suggestion and revised lines 257-261. We added a few sentences on this reviewer's requested elaboration on derivatized versus non-derivitized methods (lines 266-277)
We have changed currently to current in line 309.

Reviewer 2 Report
This is a very well-written review.
I would recommend to include figure(s) and/or table(s) showing
1) The technologies that might be used/expanded for NBS, together with the list of conditions that might be (better) screened in the near future using the different approaches (including advantages and drawbacks).
2) A summary of guidelines for NBS laboratories aiming at continuing to do good and avoid harm
Author Response
We would like to thank this reviewer for her/his valued work and compliments.
As to her/his critique we would like to state the following:
We agree that additional tables that would summarize technologies that might be used/expanded for NBS, together with the list of conditions that might be (better) screened in the near future using the different approaches (including advantages and drawbacks), would be useful, as would a summary of guidelines. That, however, would also be an almost completely new and time-consuming assignment for the authors, justifying one or more additional papers. And while we are proud of the stature of this selected group of co-authors, experts in their fields, we are not confident that we are, as a group, sufficiently positioned to issue those guidelines. Finally, professional organisations, especially ISNS, at this moment, prefer the role of trusted advisor, instead of issuing guidelines that may be interpreted as overly directive, e.g. towards governments.
Hence our statement in the manuscript:
In recent years stakeholder groups such as Screen4Rare (www.screen4rare.org) have sought to develop discussion and provide information to enable effective health policy development and inform political representatives, while recognizing and respecting national autonomy.
For all these reasons we hope that, while we fully agree with the nature of this reviewer's remarks, it is accepted that we, respectfully, do not make changes based on this valued comment.
Reviewer 3 Report
Very nice overview/update of the NBS field.
It might be worth considering a sentence or two on KREC screening (although not absolutely necessary) as it provides an added value in the genomic SCID screening (using TREC only),by identifying severe B cell deficiencies (XLA) and late onset SCID.
It might also be worth considering updating the the number of countries screening for SCID (and SMA), depending on when this manuscript will be published. It is a rapid moving field
Spelling mistake line 282.
Author Response
We would like to thank this reviewer for her/his valued work and compliments. Below, we have added a point-by-point rebuttal to her/his comments.
On behalf of the team of authors,
Sincerely yours,
Peter Schielen
We have adopted this reviewers suggestion and added a statement on XLA-screening. (line 425-428)
We acknowledge that both SMA and SCID screening are fast changing fields. Therefore the number of countries screening for SMA and SCID is a moving target. To meet this reviewers point we added lines mentioning recent introductions and including this reviewers point that SMA and SCID are dynamic fields (line 422-428). We hope that this satisfies this reviewer.
We repaired the typo in line 282